# Supercritical Fluid Gaseous and Liquid States: A Review of Experimental Results

**DOI:** 10.3390/e22040437

**Published:** 2020-04-13

**Authors:** Igor Khmelinskii, Leslie V. Woodcock

**Affiliations:** 1Department of Chemistry and Pharmacy, and CEOT, University of Algarve, 8005-139 Faro, Portugal; 2Department of Physics, University of Algarve, 8005-139 Faro, Portugal; lvwoodcock@ualg.pt

**Keywords:** gas-liquid, critical point, supercritical fluid, supercritical mesophase, universality hypothesis

## Abstract

We review the experimental evidence, from both historic and modern literature of thermodynamic properties, for the non-existence of a critical-point singularity on Gibbs density surface, for the existence of a critical density hiatus line between 2-phase coexistence, for a supercritical mesophase with the colloidal characteristics of a one-component 2-state phase, and for the percolation loci that bound the existence of gaseous and liquid states. An absence of any critical-point singularity is supported by an overwhelming body of experimental evidence dating back to the original pressure-volume-temperature (*p-V-T*) equation-of-state measurements of CO_2_ by Andrews in 1863, and extending to the present NIST-2019 Thermo-physical Properties data bank of more than 200 fluids. Historic heat capacity measurements in the 1960s that gave rise to the concept of “universality” are revisited. The only experimental evidence cited by the original protagonists of the van der Waals hypothesis, and universality theorists, is a misinterpretation of the isochoric heat capacity *C_v_*. We conclude that the body of extensive scientific experimental evidence has never supported the Andrews–van der Waals theory of continuity of liquid and gas, or the existence of a singular critical point with universal scaling properties. All available thermodynamic experimental data, including modern computer experiments, are compatible with a critical divide at *T*_c_, defined by the intersection of two percolation loci at gaseous and liquid phase bounds, and the existence of a colloid-like supercritical mesophase comprising both gaseous and liquid states.

## 1. Introduction

In 1863 J. W. Gibbs introduced the concept of a thermodynamic state function to describe physical and chemical equilibrium states. Gibbs treatise [1] first explained how various thermodynamic physical states of a pure fluid are defined as points on surfaces. Any state, for example, of a one-component (*C* = 1) fluid volume pressure-temperature surface *V*(*p*,*T*), as illustrated in Figure 1, can be defined by the intersection of two lines. A state with two degrees of freedom (*F* = 2) is defined by the intersection of an isotherm and an isobar. For states with 2 phases (*P* = 2) where *F* = 1, either an isotherm or an isobar crosses a coexistence line. A coexistence line can be defined where two lines cross on Gibbs chemical potential surface. The triple point, whereupon *F* = 0, is where two coexistence lines cross in the (*p,T*) plane, etc. 

In addition, shown in Figure 1 is a point, labelled “critical state”, which has not been defined thermodynamically, i.e., by intersection of state function loci. At a critical temperature *T*_c_, there is deemed to be a merging of the properties of the gas and liquid states in coexistence. At a critical temperature *T*_c_, Andrews proposed the hypothesis of “continuity of liquid and gas” [2]. He suggested that there is no distinction between gas and liquid at or above *T*_c_ and hence definitive state bounds along any supercritical isotherm. The experimental pressure measurements for CO_2_, as a function of density and temperature, in the original paper by Andrews are shown in Figure 2a.

The first attempt to explain Andrews’ experimental results quantitatively was by van der Waals. He published his renowned thesis [3] with essentially the same title translated into Dutch as Andrews’ original paper. The van der Waals thesis, however, is mainly about a hypothesis regarding a proposed cubic *p*(*V*,*T*) equation-of-state that predicts a singular point on the Gibbs density surface. The equation of van der Waals is continuous in density at *T*_c_ whereupon the first two derivatives of pressure with respect to changes in density or volume go to zero at a singular node on the Gibbs density surface. At this hypothetical point, the difference in density, and hence all other distinguishing properties, between gas and liquid are deemed to disappear. For nigh 150 years, Andrews–van der Waals theory of critical and supercritical continuity of gas and liquid has been the accepted and taught physics of liquid-gas criticality. The critical point envisaged by van der Waals, however, has never been defined thermodynamically, and has always remained an experimentally unsubstantiated hypothesis. 

Alongside Andrew’s original data for CO_2_, we show in Figure 2b the same 6 isotherms taken from a modern experimental *p-**ρ**-T* thermodynamic fluid property databank [4]. Also shown are the coexistence curves and the law of rectilinear diameters (LRD) extended into the supercritical region [5]. A foremost observation is that, when the density is on a linear scale, there is a linear region to the supercritical isotherms centred upon the extended LRD. The slope $\omega_{T}={\left(\partial p/\partial \rho \right)}_{T}$, i.e., the thermodynamic state function rigidity, decreases with T at this midpoint, approaching zero at the critical temperature (*T*_c_ = 29.9 °C). 

Note also that ${\left(\partial p/\partial \rho \right)}_{T}$ decreases with density for gas isotherms and increases with density for liquid. A distinction between subcritical coexisting gas and liquid phases evidently extends to the supercritical isotherms. This fundamental difference between gas and liquid has not received the observation and attention it deserves. The practice of plotting pressure-volume, instead of the natural intensive variable *density*, can obscure to some degree the linear region of the supercritical fluid phase. In addition, the commonly used compressibility $-\frac{1}{V}{\left(\frac{\partial V}{\partial p}\right)}_{T}$, a mechanical property used by van der Waals, and by the science community since, is rather uninformative. By contrast, its reciprocal, rigidity $\omega_{T}={\left(\partial p/\partial \rho \right)}_{T}$, with dimensions of molar energy, is more amenable to molecular interpretation. Indeed, it is the thermodynamic state function defined as the work to increase the density of the fluid. Rigidity is directly proportional to the reciprocal molecular number density fluctuations. 

Had Andrews published his data with a linear density scale, early indications of non-continuity between gas and liquid might have attracted more attention. Moreover, in 150 years since Andrews’ discovery of a critical temperature (*T*_c_), nobody has reported a direct observation of a liquid-gas critical density. This hypothetical property has only ever been defined and obtained by an a priori assumption of its existence by application of LRD [5], which seems to be illogical cyclic reasoning. 

Soon after van der Waals suggested his 2-parameter equation-of-state, it was found to be inadequate for the *p*-*V*-*T* data in the critical, near-critical and supercritical region. Empirically parameterized cubic equations-of-state, in order to be accurate in the critical region, have required ever-increasing numbers of adjustable parameters to reproduce the thermodynamic state functions with 5- or 6-figure experimental numerical accuracy. Current research and modern compilations require 25+ parameters [4]. This presents a typical case of Bayesian adjustment of the hypothesis, which may be an indicator of faulty reasoning. Thus, this progressive inadequacy of cubic equations of state is per se indicative that the hypothesis of supercritical continuity is fundamentally incorrect. There are many other indicators from experimental sources that have been available all along but have been ignored. In sections II to IV below, we review a selection of the salient research reports that have provided compelling experimental evidence for the non-existence of van der Waals critical volume point. 

Computer experiments by molecular dynamics (MD) simulations and Monte Carlo (MC) methods became a research tool in the latter part of last century. Computer experiments have certain advantages over real laboratory experiments, at least the absence of impurities and the absence of gravity. The evidence from computer experiments relies on the usage of simple model molecular Hamiltonians, ranging from sticky spheres, square-well models, including the van der Waals mean-field limit, to Lennard–Jones fluids. All of these model classical fluids, without exception, show an unequivocal connection between percolation transitions that characterize the limits of gas and liquid phases, and criticality. The percolation transition that bounds the gas phase (designated PB) occurs at the density whereupon the size of bonded molecular clusters suddenly diverges to length scales of the order of the size of the system. The percolation transition that bounds the liquid state (PA) is when available volume pockets in the liquid, large enough to accommodate an additional molecule, also suddenly diverge. The critical point (*p*_c_, *T*_c_), on Gibbs *p*-*T* surface, is the intersection of the two percolation loci, which bound the existence of pure gas and liquid states [6,7,8]. At this juncture, the two states, liquid and gas, have the same *T*, *p* and chemical potential, but slightly different densities, by about 20%.

Considering this body of empirical evidence, we ask how could the erroneous theory of liquid-gas criticality, which is so clearly inconsistent with the experimental literature, have survived for so long? One reason is the misguided generalization of Onsager’s two-dimensional (2D) Ising model solution [9]. This mathematical tour-de-force also, eventually, gave rise to the concept of ‘universality’ first mooted by Uhlenbeck [10], yet accepted as established science by Rowlinson [11], Fisher [12] and Sengers [13], and further extended by Kadanoff and his collaborators [14]. The concept transcending dimensions 1D to 3D was later supported in the three papers on the Kac mean-field model in 1, 2 and 3D [15,16,17] and renormalization group theory of Wilson [18]. However, universality is inconsistent with our knowledge of percolation transition loci, and their relationships to criticality and supercritical properties of the liquid and gaseous states. In fact, percolation phenomena are non-existent in 1D, and very different between 2D and 3D, thereby invalidating the generalizations of the universality hypothesis. 

We review below the experimental evidence, spanning 150 years of research, which indicates the existence of a supercritical mesophase, neither liquid nor gas, but a colloidal dispersion in which both gaseous and liquid states can percolate the phase volume [19]. This could not happen in 2D as percolations must coincide. This colloidal mesophase is a new macroscopically homogeneous equilibrium state of matter besides crystal, liquid and gas. It is a single homogeneous Gibbs phase, with two degrees of freedom, but which is inhomogeneous on the nanoscale. The experimental evidence for its existence has been in the literature since the early 1900s, i.e., two years before van der Waals was awarded his Nobel Prize in 1910 for his thesis “On the continuity of gas and liquid states” [3]. 

## 2. Experimental Gas-Liquid Equilibria

### 2.1. Computer Experiments

The supercritical mesophase was discovered originally from computer experiments on model square-well (SW) fluids, which can be understood as a perturbation of the hard-sphere (HS) fluid. The hard-sphere fluid exhibits two percolation transitions of the excluded volume (PE) and accessible volume (PA), between which both complementary volumes (*V*_A_ + *V*_E_ = *V*) percolate the phase volume. Although the HS fluid has a well-defined mesophase, it cannot account for gas-liquid condensation and 2-phase coexistence without an attractive potential term. The SW-model molecular Hamiltonian is defined by adding a constant energy of attraction (*ε*) of finite width (*λ*) to a HS (diameter *σ*) pair potential. The range must be finite, and not infinite, as implicit in the attractive term of van der Waals equation and similar mean-field theories, to give a fluid that complies with thermodynamic laws. The SW attraction introduces another percolation transition of cohesive gaseous clusters, the density of which depends upon the range of the attraction *λ*, as also shown in Figure 3. Properties of square-well fluids of various values of well-width *λ* have been reported from computer experiments. The simulation data shown in Figure 3 are taken from references [6,7,20,21,22,23,24,25]. These results from computer experiments are reproduced from the original reports of the existence of the supercritical mesophase [6,7]. 

The effect of the square-well perturbation on the HS fluid properties is to reduce the pressure difference between the two percolation transitions PE (or bonded-cluster percolation PB for *λ* < 2*σ*) and PA, because there is a much higher cohesion with decreasing temperature at the higher liquid density. The temperature dependence of the percolation transitions is only slight [11]. At a critical temperature (*T_c_*) the two percolation transitions intersect, the pressures and hence the chemical potentials become equal. At the critical percolation intersection state point, *T_c_*-*p_c_*, there is a maximum coexisting gas-phase density, and a minimum coexisting liquid-phase density. 

The hard-sphere mesophase at high temperature or large well-width can be seen to narrow to around 20% in the sticky-sphere limit *λ* → 0. Computer experimental results show maximum observable coexisting vapour phase densities and minimum coexisting liquid phase densities. Viewed alongside the HS percolation transitions, these results are inconsistent with the theoretical concept of a singular critical density with divergent thermodynamic properties. The observation of a critical divide, and its empirical percolation intercept definition origins, with PB and PA boundaries, extending all the way to the HS limit (*k*_B_*T*/*ε* → infinity) for any value of well-width *λ*, defines the existence of a supercritical mesophase. 

A critical pressure along any isotherm is defined by the intersection of the two percolation transitions. Above *T_c_*, the liquid percolation has the higher pressure, below *T_c_* the (metastable) liquid percolation loci, previously known as spinodal, have the lower pressure. At the intersection, the two states, gas and liquid, have the same *T* and *p*, and hence also the same *μ* (chemical potential), but different densities. Two coexisting states with constant *T*,*p*,*μ* at the percolation loci intersection define the critical point *T_c_*-*p_c_* in the *T*-*p* plane, and a line of critical transitions from 2-phase coexistence (P = 2, F =1) to a single phase (P =1, F = 2), for *T* > *T_c_*.

Since the original discovery [6,7] of the mesophase in the square-well fluids, equally compelling empirical evidence for the supercritical divide and a supercritical mesophase have been reported for other model molecular Hamiltonians, with essentially the same conclusions. An extensive computer simulation study of the Lennard–Jones (L-J) fluid [25] reported a critical density hiatus and a supercritical mesophase with linear isothermal state functions of density. Note that there are no impurities or gravitational potential in computer experiments! The pressure-density data points were computed for systems of 10,000 and 32,000 L-J atoms for 2000 state points along 6 near-critical isotherms with the results shown in Figure 4. The critical isotherm shows a perfectly horizontal straight line between coexisting maximum gas and minimum liquid densities. Significantly, there was no observable particle number dependence that one might expect to see in the vicinity of any critical singularity with divergent properties. 

Another advantage of computer simulation is that detailed structural data can be obtained that helps to explain the phenomenological behaviour at the molecular level. The Lennard–Jones fluid was found to have a critical isotherm at *kT_c_*/ε = 1.3365 ± 0.0005 (where *k* is Boltzmann’s constant and *ε* is the attractive minimum energy of the L-J pair potential). Several supercritical isotherms were studied in more detail with a very high precision, including the *kT*/ε = 1.5 isotherm over the whole density range. Accurate structural information was obtained, including the radial distribution functions (RDFs) over the whole density range. Although individual state point RDFs are of very high statistical precision, they do not reveal any information about subtle structural changes that must inevitably accompany the onset of hetero-phase fluctuations, and the higher-order order percolation transitions. When the normalized pair probabilities *g*(*r*) at specific interatomic distance (*r*) are viewed as a function of density, however, a picture of the three supercritical regions, with the mesophase, emerges. The detailed density-dependent structural properties (Figure 5) show the unequivocal evidence that there exist three distinct structural state regions, liquid, meso and gas. The *g*(*r*,*ρ*) also reveals the evidence of the hetero-phase fluctuations that are the precursor states to the percolation transitions. The structural data reproduced in Figure 5 show that at all intermolecular distances from highly repulsive overlap to long-range distances of four times the most probable pair distance, there are extremely slight, but statistically significant, structural differences, between gas, liquid and meso, as evidenced by the subtle deviations from uniform probabilities as a function of density along a supercritical isotherm. 

### 2.2. Surface Tension

Thermodynamic equilibrium requires the surface tension of a vapour in coexistence with a liquid at subcritical temperatures to be positive. The first-order phase transition is characterized by a 2-phase coexistence region with a latent heat, molar enthalpy change at constant temperature and a change in molar volume at constant pressure. If the surface tension, and hence also interfacial excess Gibbs energy, were to be zero or negative, first-order coexistence with phase separation would not be possible as the two states would spontaneously inter-disperse. This is what happens at, and above, the critical temperature between the limits of existence of gas and liquid states on a *p*-*T* density surface.

This general phenomenology of a critical density hiatus can be further understood by considering the role of surface tension. If the percolation locus PA is the boundary of the existence of the gas or liquid states for supercritical temperatures, it must connect up with the boundary for the non-existence of the metastable gaseous and liquid states for sub-critical temperature, i.e., the vapour or liquid spontaneous decomposition spinodals. The spinodal lines are often defined operationally by the absence of a barrier to nucleation of the new phase. An alternative definition, however, is the point at which the surface tension of metastable supersaturated liquid-gas goes to zero, as suggested by He and Attard [26]. 

At *T*_c_, when the percolation lines PA and PB intersect, there is no barrier to nucleation; hence the surface tension must go to zero at the different coexisting respective densities of gas and liquid. Evidence that this indeed happens can be found in a Monte Carlo computer calculation of the surface tension of Lennard–Jones fluids by Potoff and Panagiotopoulos [27]. In the original interpretation of their MC results, these authors overlooked the fact that the surface tension becomes zero at a finite density difference, having a priori assumed the van der Waals and universal singularity hypotheses to be established science. 

In Figure 6, using an asterisk superscript to denote L-J molecular-reduced dimensionless values, we replot the numerical MC surface tension data in the vicinity of *T*_c_. It varies with the cube of density difference between gas and liquid, and becomes zero, not at a van der Waals hypothetical critical point at which the density difference is zero, but at a finite density difference. 

There are two other noteworthy aspects of the data by Potoff and Panatagiopoulos [27] seen in Figure 6. First, the data point at *T** = 1.310 has a density difference Δ*ρ** = 0.165. At the next data point, which is defined as *T**_c_ = 1.312, Δ*ρ** = zero, and γ* is reported to be very slightly negative. Zero or negative surface tension would be consistent with the critical dividing line above which there is a mesophase of a colloidal nature. On the liquidus side, above *T*_c_, if the surface tension of a real fluid were to be zero or negative, there will be no barrier to the spontaneous fluctuations to gaseous states, which percolate the phase volume below *ρ*_PA_. Likewise, on the gas side, for densities above *ρ*_PB_, there will be no barrier to the spontaneous formation of percolating liquid-like droplets. Secondly, the data from reference [27] shows surface tension *γ** varying linearly with *T*. We obtain from Table 1 in reference [27] *T** = 2.5975*γ** + 1.3093. It follows that the density difference *ρ*^*^_liq_ – *ρ*^*^_gas_ is a linear function of (*T* – *T*_c_)^1/3^ in the immediate vicinity of *T*_c_ as seen in Figure 7. 

In the coexistence region, the thermodynamic property that also characterizes the difference between gas and liquid states in coexistence is the surface tension (*γ*). It is simply defined as the reversible work or Gibbs energy, required to create an area (*A*) of plane interface between the two coexisting phases with the same *T*, *p* and *μ*, i.e., *G*_s_ = *A*γ. For all subcritical gas-liquid coexisting states in the 2-phase region both *G*_s_ and *γ* are positive, whence *G*_s_ and therefore *G*_total_ for the whole system is minimized by phase separation. For the equilibrium liquid in coexistence however, from the triple point to the critical point, the surface tension goes to zero. The difference between the rigidities of coexisting liquid and gaseous states (Δ*ω* = *ω*_liq_ − *ω*_gas_) correlates with the surface tension (Figure 6). This can be explained: the difference in work required to increase the density of the coexisting states by adding molecules at constant volume is physically equivalent to the work required to create the interface by reducing the density of the liquid and simultaneously increasing the density of the vapour on either side of the Gibbs dividing surface; a line defined within the interface such that the total mass within an arbitrary volume containing the interface remains constant. 

Thus, although the surface tension and the rigidity difference correlate, both approaching zero at *T*_c_ (Figure 7), the density difference of the coexisting homo-phases, both at *T*_c_ and in the supercritical range *T* > *T*_c_, must remain finite. This suggests that the critical state of two coexisting homo-states within a single Gibbs phase extends well into the supercritical region, and probably all the way to the Boyle temperature as evidenced by the correlation of equation-of-state thermodynamic properties [28]. In the next section, we will re-assess the experimental evidence from historic laboratory measurements of near-critical and supercritical thermodynamic properties in the new light of the supercritical mesophase discovered in computer experiments. 

### 2.3. Mesophase Dispersion

A colloidal dispersion is defined as one state, solid, liquid or gas, dispersed as small micro-scale particles in a continuous state, also solid, liquid or gas. All well-characterised colloids hitherto comprise at least 2 distinct chemical components, e.g., an emulsion of water in oil, or foam of air in water. A dispersion of liquid in gas is a “mist”. We will see from the evidence reported below that there is now every reason to believe that a fourth equilibrium state of a pure one-component thermodynamic fluid system exists. The pre-percolation state can be considered a colloidal liquid-in-gas dispersion with the liquid state dispersed in the gas phase as very large clusters or hetero-phase fluctuations, or gas-in-liquid vice versa. Within the mesophase both states percolate the phase volume. 

The earliest experimental evidence for the existence of dispersed liquid at temperatures above *T*_c_ can be found in the papers by Bradley et al. in 1908 [29]. Notwithstanding these results, in 1910 van der Waals gave his Nobel lecture [30] in which he ignored the Bradley et al. research with the declaration “at the critical volume the densities of gas and liquid become equal”. After exploring the effect of mechanical vibrations on the properties of near-critical CO_2_, however, Bradley et al. were able to state with some certainty that their experiments “proved” the existence of two phases far above the critical point. Moreover, their experimental observation that the meniscus disappears over a range of mean densities has never been properly explained. This reproducible experimental result, reviewed by Mills [31], was dismissed as an artefact of gravity, impurities or non-equilibrium states by the Dutch school in their argument in favour of verification of van der Waals hypothesis, and generally ignored by many of his disciples, but with the notable exception of Traube [32] who cited 10 experimental papers published prior to 1908 that vitiated the van der Waals hypothesis. 

The observations of Traube are essentially correct, because, as we now know, the densities do not become equal at *T*_c_. Both liquid and vapour coexisting densities at *T*_c_ are bounded by the respective percolation loci. At the temperature of intersection of the percolation loci (*T*_c_), the meniscus disappears within the tube over a certain range of mean densities, even when the intensive thermodynamic state functions, including the densities, of coexisting liquid and gas phases, are still unequal. 

Since the earliest *p*-*V*-*T* measurements of Andrews [2], no significant measurements were reported until Maass and co-workers during the 1930s [33,34,35,36,37,38,39,40,41]. Using a method of direct local density probes, with a spring and float arrangement, Maass et al. found many significant new effects in the critical region of fluids. They observed hysteresis effects on repeated passage through the critical point: equilibration over time was very important. The last line of their conclusion section at the end of their first paper [33] is, quote: “briefly, it may be stated that the results cast considerable doubt upon van der Waals classic theory of the continuity of state”. In the one-phase region above *T*_c_, Maass et al. measured large density differences and density gradients at fixed temperature and pressure. The description of their findings and even the titles of some of their publications, such as “Persistence of the Liquid State of Aggregation above the Critical Temperature” are inconsistent with the van der Waals hypothesis. Maass and co-workers acknowledged [35,36,37] that some of these effects had been seen before and gave due recognition to Traube and his co-researchers who had reached essentially the same conclusion [32]. 

Traube defined the critical temperature as that at which the two solubility curves (gas in liquid and liquid in gas) intersect. This insight by Traube is remarkably close to the interpretation of criticality based upon the intersection of the two percolation loci that bound the supercritical liquid and gaseous states [7,8,9]. 

Maass and co-workers found the supercritical fluid region to be a dispersion of a liquid in a vapour, in contrast to the idea of continuity of states, and were able to conclude that “a liquid-like structure persisted above *T*_c_.” These conclusions, from nearly a decade of painstaking research by Maass and his collaborators, are compelling evidence for the existence of a supercritical mesophase. In their last paper in the series Maass and Noldrett [39] write, quote: “Further evidence of the existence of a two-phase system above the temperature of meniscus disappearance as normally determined is presented. The existence of a latent heat of vaporization above this temperature is pointed out”. Finally, Maass et al. in the last articles [40,41] conclude “this is considered to be evidence for the formation of a dispersion of liquid and vapour before the critical temperature is reached”. Maass et al. suggested the nomenclature "critical dispersion temperature" instead of critical point [41]. 

### 2.4. Density Hiatus

The experimental situation in 1945 is summarized by a plot of data in the classic paper on corresponding states by Guggenheim [42] (Figure 8), which shows that the fundamental coexistence properties of simple atomic and molecular liquids have the same form on reduced scale. This collection of experimental points quite clearly shows a lower density above which there are no coexisting gas phase density measurements, and a higher density below which there are no coexisting liquid phase measurements. Since there is no satisfactory scientific explanation for this hiatus, this itself is *prima facie* evidence for the non-existence of van der Waals continuity of liquid and gas and the non-existence of a critical point on Gibbs density surface. Ever since Guggenheim’s paper in 1945, moreover, no experimental observations of gas or liquid states of coexisting densities within the critical divide have ever been reported. In fact, the same also applies to computer studies of liquid-gas critical coexistence densities. 

Of special interest in Guggenheim’s experimental data analysis is the observation that fluids neon, argon, krypton and xenon, all exhibit the same phenomena with corresponding states of maximum coexisting gaseous density states, and corresponding minimum coexisting liquid density states. This is further evidence that there is no significant gravitational mass effect on critical properties. Guggenheim’s plot also shows molecular liquids oxygen, nitrogen, carbon dioxide and methane exhibiting the same critical phenomenology. The principle of corresponding states that actually originated in van der Waals hypothesis, therefore, will still hold for the revised description of gas-liquid criticality if properties are reduced phenomenologically using *p*_c_, *T*_c_ and a mid-critical density *ρ*_c_ defined by the law of rectilinear diameters. Likewise, any molecular-reduced corresponding states law also still applies, using molecular size, molecular mass and a characteristic interaction energy to define reduced time. The revised science of criticality does not change scaling effects of size or mass of a molecule, on thermodynamic state functions. 

In the early 50s, the experimental investigations into the dilemma were resumed in the laboratory of Schneider [43,44,45]. Measurements of the *p*-*V* isotherms for sulphur hexafluoride in a small temperature range of approximately 1.5 °C in the critical region were first reported [43]. All *p*(*V*)*_T_* isotherms above *T*_c_ were found to have a finite slope, but it was not possible to say whether the critical isotherm has a finite horizontal portion or "flat top". Following further experiments on xenon, Weinburger and Schneider [44] found a coexisting density gap but argued that the van der Waals theory can explain a large portion of the flat top if the effect of gravity is taken into account. Initially, Schneider and his co-authors had reported a flat top but also inferred, erroneously as we shall see below, that it was the consequence of gravity. 

After further refinement of the experimental techniques, however, Hapgood and Schneider [45] reported further *p*-*V*-*T* measurements of xenon, extending from several degrees above *T*_c_, to below the *T*_c_, and at densities starting well above the critical density. They concluded: “the critical isotherm is considerably flatter and broader over a range of densities than that corresponding to a van der Waals equation, and at the critical point, the third and fourth derivatives of pressure with respect to volume also appear to be zero”. This had been the conclusion of two independent theoretical predictions, just two years before Hapgood and Schneider’s paper, by Zimm [46] and by Harrison and Mayer [47]. Zimm found that all of the derivatives at *T*_c_ are zero. Any state function for which all the derivatives go to zero is a horizontal line [45]. There followed theoretical papers by O.K. Rice [48,49] supporting the conclusions of Zimm, Mayer and Harrison of the existence of a density hiatus at *T*_c_. 

Thus, by the end of 1950s, van der Waals hypothesis had been discredited to a large extent both by experiment, and the by the Mayer cluster-integral formalism, which is formally correct for a two-body Hamiltonian model, albeit analytically intractable. By this time Onsager’s analytic solution of 2D-Ising model partition function [9], however, was giving rise to a new era of theoretical physics of phase transitions in lattice gases, and eventually to the concept of universality [10].

### 2.5. Percolation State Bounds

To demonstrate that the percolation state bounds, which can be clearly defined and seen in computer experiments [6,7], exist also in real fluids, we only need to consider one such fluid as a test case. Take, for example, the experimental measurements from Gilgen et al. for argon [50,51]. The data points in Figure 9 are plotted directly from the published data in reference [50]. Except in the immediate vicinity of the critical temperature (i.e., within 0.1K), these data are reported with six-figure accuracy. The loci of the two percolation transitions are obtained from the discontinuity in the slope of the *p*(*ρ*) isotherms, for the seven supercritical isotherms shown. The percolation transitions define the bounds of existence of the supercritical gas and liquid phases. The critical isotherm is horizontal between the maximum gas density of coexistence and the minimum liquid density at *T*_c_. 

Neither of the percolation bounds (PA gas and PB liquid [8]) are extrapolations of the coexistence line. At *T*_c_, both PA and PB (Figure 10) intersect in the *p-T* plane and cross the critical coexistence line, to become subcritical limits of existence of the metastable saturated compressed gas (*p* > *p*_sat_) and expanded saturated liquid phases (*p* < *p*_sat_), respectively. These percolation loci define the limits of existence of gas and liquid phases, not only above *T*_c_, i.e., in the supercritical region, but also in the subcritical metastable regions of existence of gas and liquid states. 

Nobody has ever reported a direct observation of a critical density [5]; this is well illustrated by the extensive high-precision experimental measurements of the argon liquid-vapour coexistence densities by Gilgen et al. [51]. The highest temperature for which they report both coexisting vapour and liquid densities is 150.61 K. They use a cubic scaling equation and a law of rectilinear diameters to obtain their critical point temperature (150.69 K), and critical density of 535.6 kg/m^3^. The mean of the two extreme recorded liquid and vapour densities is 536 kg/m^3^. The lowest co-existing liquid mass density they report is 602 kg/m^3^. The highest vapour mass density they can observe near *T*_c_ is 470 kg/m^3^. The line of critical states connects these two points (Figure 9). 

## 3. Thermophysical Property Compilations

### 3.1. Multiparameter Equations-of-State

There is now an abundance of highly accurate *p-V-T* experimental measurements for a diversity of atomic and molecular fluids. The most prolific contributors to the NIST databank [4] are the research group of Prof. W. Wagner. Over many decades, they have performed a great service in high-precision measurement and presentation of invaluable thermodynamic data for many academically and industrially important fluids, as evidenced by NIST physical property databank citations. Experimental researchers since 1965 have never questioned the authenticity of van der Waals hypothesis as “accepted science”, propagated since the mid-1960s via the concept of universality. The Wagner group equations-of-state are not able to reproduce with normal precision the near-critical properties in most of their reports. This admission is now entirely understandable. It is consistent with the non-existence of a singularity, and yet emphasizes what meticulous service the Wagner group have provided to the engineering and physical science community in awkward circumstances [4]. 

From the modern literature of thermodynamic *p-V-T* data on atomic and molecular fluids, one could choose any one of the 200 fluids listed in the NIST thermo-physical data bank or any one of the hundreds of experimental papers in the literature. Numerical equations-of-state from the literature, however, should also be viewed with circumspection, especially in the vicinity of the critical temperature. The numerical representations of raw experimentally measured data points [4] used in parameterizations are predetermined by an a priori assumption at the outset of the existence of a van der Waals singularity at a critical density, and continuity of the supercritical equation-of-state in all its derivatives. The so-called universal exponents for the description of the thermodynamic properties in the immediate vicinity of *T*_c_ have also been incorrectly employed. Thus, even though no one has ever measured a vanishing density difference up to *T*_c_ directly, all the NIST thermodynamic state functions have mistakenly presumed a continuity of gas and liquid to be the underlying science from the outset.

One reason for the inadequacy of complex multiparameter equations-of-state with increased experimental precision is that the continuous functional forms are fundamentally incorrect in the vicinity of *T*_c_ and in the supercritical mid-range between gas and liquid phases. The mesophase, confined within percolation loci that bound the gas and liquid phases by higher-order discontinuities, can readily be identified. A simple numerical differentiation of NIST equations-of-state, for example, can demonstrate the existence of the supercritical mesophase and locate the phase bounds, along any isotherm, of any fluid (e.g., CO_2_ at *T*/*T*_c_ = 1.25) for any of the 200 fluids in the NIST Thermophysical Property data bank [4]. These boundaries are smoothed over by the equations-of-state used to parameterize the original experimental data. 

### 3.2. Rigidity Symmetry at State Bounds

The literature *p-V-T* data show inequalities that distinguish gas from liquid in the supercritical region, and bound the phases according to the respective percolation loci PB (bonded cluster of gas molecules) and PA (available volume or voids in liquid state) which evidently coincide with discontinuities, probably appearing in the third derivatives of Gibbs energy with *p* or *T*. Rigidity (*w*) is the work required to isothermally increase the density of a fluid; with dimensions of a molar energy. This simple state function relates directly to the change in Gibbs energy (*G*) with density at constant *T* according to: (1)$$\omega_{T}={\left(\frac{\partial p}{\partial \rho }\right)}_{T}=\rho {\left(\frac{\partial G}{\partial \rho }\right)}_{T}$$
showing that the rigidity must always be positive. Gibbs energy cannot decrease with pressure when *T* is constant. From these definitions, moreover, not only can there be no continuity of gas and liquid, but the gas and liquid states are fundamentally different in their thermodynamic description. For all gaseous states below the Boyle temperature (*T_B_*) rigidity decreases with density:
(2)$$\mathrm{Gas}:\;\rho \;<\;\rho_{\mathrm{PB}},\;{\left(\frac{\partial \omega }{\partial \rho }\right)}_{T}<0$$

For a high-density state, liquid, rigidity increases with density: (3)$$\mathrm{Liquid}:\;\rho \;>\;\rho_{\mathrm{PA}},\;{\left(\frac{\partial \omega }{\partial \rho }\right)}_{T}>0$$

In the mesophase the rigidity is constant:
(4)$$\mathrm{Mesophase}:\rho_{\mathrm{PB}}\;<\;\rho \;<\;\rho_{\mathrm{PA}},\;{\left(\frac{\partial \omega }{\partial \rho }\right)}_{T}=0$$

This discontinuity between the gas and liquid states in the supercritical region of all 200 of the NIST atomic and molecular fluids is the irrefutable experimental evidence against any critical and supercritical continuity of liquid and gas (see Figure 11 for argon). There is a hiatus, the mesophase, within which the extensive thermodynamic properties, and notably density *ρ*(*p*,*T*), are found to obey a linear equation-of-state in this region, suggesting a supercritical mesophase linear combination similar to the subcritical lever rule. 

Rigidity is determined by number density fluctuations at the molecular level, which have different but complementary origins in each phase, hence there is certain symmetry between liquid and gas along the same isotherm on either side of the critical divide. The origins of this symmetry are many small clusters in a gas with one large void, and many vacant pockets in the liquid with one large cluster. An occupied site and an unoccupied void have the same statistical distribution of local structural properties, which stems from a molecular definition of chemical potential as the probability of adding one more molecule to an equilibrated fluid [52]. 

### 3.3. Physical-Constant Equations-of-State 

In order to demonstrate a necessity of three separate equations-of-state, we take, as an example, a supercritical isotherm *T* = 350 K, or *T*/*T*_c_ = 1.15, of CO_2_. The critical and Boyle temperatures for CO_2_ according to NIST databank [4] are *T*_c_ = 305 K and *T*_B_ = 725 K. Using the same method described previously for argon [8], the coexisting densities at *T*_c_ are found to be *ρ*_c_(gas) = 7.771 mol/L and *ρ*_c_(liq) = 13.46 mol/L. Substituting these physical constant values into equations (5 to 7) for *T* = 350 K, we obtain the gas and liquid state density bounds *ρ*_PB_(gas) = 7.305 mol/L and *ρ*_PA_(liq) = 12.02 mol/L, and the rigidity *w*(350 K) = 0.318*kT* (0.925 kJ/mol). 

If we try to fit the NIST data for the whole isotherm continuously from 0, just up to only 50 MPa, using a 6-term polynomial for example, we obtain the result:

*p* = –0.000008*ρ*^6^ + 0.000457*ρ*^5^ – 0.009373*ρ*^4^ + 0.096245*ρ*^3^ – 0.596317*ρ*^2^ + 3.459264*ρ*– 0.189239 with mean square regression *R*² = 0.999125. 

In this typical continuous overall polynomial fit, the lower-order coefficients are scientifically meaningless. In contrast, by using virial coefficients and physical properties of the mesophase bounds, the NIST data for the whole range of three regions can be reproduced with essentially 100% precision when the gas, liquid and meso-states have different equations-of-state. We assume that there is at least a third-order phase transition, i.e., a discontinuity in the third derivative of Gibbs energy at the gas and liquid-state boundaries in accord with inequalities (2 and 3). 

The experimental data from the NIST tabulations are reproduced in Figure 12 with the same precision as the original data can be obtained, i.e., accurate to 5 or 6 figures, as evidenced by the mean-squared regression (*R*^2^) in the trendline polynomial coefficients as given. In addition, shown in Figure 12 are redefined equations to give the virial coefficients as shown on the plots. The coefficient *a*_1_ in the liquid-state expansion is taken to be equal to the value of *w_T_* in the mesophase to describe the third-order discontinuity between the mesophase and the liquid state. There is a symmetry between the rigidity of gas and liquid on either side of the mesophase. This observation is empirical though it has a molecular origin in the fluctuations of the available volume and its relationship to the chemical potential of both gas and liquid states [52]. 

The rigidity isotherm data for CO_2_ (Figure 13) show a clear symmetry on either side of the mesophase along an isotherm [53], which further suggests that the mesophase bounds narrow with increasing *T* and merge at or close to the Boyle temperature *T*_B_. This is the temperature above which the second virial coefficient is positive and below which it is negative but the analytic form as it passes through zero remains unknown. The same behaviour is seen in the rigidity plots for other liquids, for example water [54]. 

Equation-of-state experimental data, albeit accurate and painstakingly obtained, may not be the most reliable to decide the issue of critical flatness. It is not easy to distinguish a low curvature region from one that is in fact a straight line in *p*(*ρ*)*_T_*. The literature critical-point universality theory predicts that the temperature or pressure scales as Δ*ρ*^d^ along the critical isotherm, which could therefore appear to be very flat anyway within the hypothesis. Many *p*-*V*-*T* experimental results for real molecular fluids have required exponents d = 3 to 4, or even higher [54] in order to parameterize an apparent flat top within the experimental uncertainty. The presentation of near-critical experimental results, however, has been adversely affected by hypotheses, which are here seen to be incorrect in the light of experimental data, and hence misrepresent the critical divide at *T*_c_ and the supercritical mesophase.

## 4. Near-Critical Heat Capacities

### 4.1. Universality and C_v_

At a time between the Rice review in 1955 [48] and the Washington Conference in 1965 [10,11,12,13], the dual hypotheses of (i) a gas-liquid critical point singularity, and (ii) super-critical continuity of gas and liquid states, evidently became accepted science. In trying to ascertain the originating argument for this acceptance, we note that in the Proceedings of the Washington Conference, for example, in both the papers by Rowlinson [11] and by Fisher [12], a critical density singularity with supercritical continuity was *a priori* assumed. An experimental review paper on transport coefficients near the critical point, by proceedings editor J. V. Sengers [13], contained no fewer than 20 mentions of “*the critical isochore*”. Nowhere in any of the papers in the entire Washington 1965 Conference proceedings, can any mention or reference be found to any of the above experimental *p*-*V*-*T* evidence, references [32,33,34,35,36,37,38,39,40,41,42,43,44,45], contradicting the van der Waals hypothesis of the critical isochore singularity. 

In an argument against an alternative thermodynamic description of liquid-gas critical and supercritical fluid properties in which there is no singularity [6,7,8], Sengers and Anismov [55] have recently resurrected historic experimental measurements of heat capacities. The results they rely upon, in support of universality concept, were the laboratory experimental reports of a divergent isochoric heat capacity of argon [56,57,58] along a near-critical isochore in the 1960s and early 1970s; and a similar NASA space shuttle microgravity measurement of the same heat capacity on sulphur hexafluoride (SF_6_) in the late 1990s [59]. 

The first and second law of thermodynamics can be simply stated: “*Q*_rev_ and *Q*_rev_/*T*, enthalpy and entropy, respectively, are state functions”; where *Q*_rev_ is the reversible heat absorbed. It follows that one cannot reversibly add heat to a classical Gibbs thermodynamic fluid without either doing work of expansion or increasing the temperature. The apparent divergence of *C_v_* in these experiments is based upon a misinterpretation of experimental heat capacities in the two-phase regions [60]. 

### 4.2. Heat Capacity Definitions

Thermodynamic state functions, internal energy (*U*) and enthalpy (*H*), and hence also the heat capacities *C_v_* and *C_p_*, are defined for the two-phase region, i.e., at *T* < *T*_c_, according to the lever rule: (5)$$C_{v}={\left(\frac{\partial U}{\partial T}\right)}_{V}=xC_{v}(liq)+(1-x)C_{v}(gas)$$
(6)$$C_{p}={\left(\frac{\partial H}{\partial T}\right)}_{p}=xC_{p}(liq)+(1-x)C_{p}(gas)$$
were *x* is the temperature-dependent mole fraction of liquid defined by the experimental density and coexisting densities: (7)$$x(T)=\frac{\rho_{liq}\left(\rho_{}-\rho_{gas}\right)}{\rho_{}\left(\rho_{liq}-\rho_{gas}\right)}$$

Whereas *C_v_* does not diverge either below or above *T_c_*, the heat capacity at constant pressure *C_p_* diverges both in the two-phase region and in the supercritical region, i.e., as *T* → *T*_c_ both above and below *T*_c_, we obtain:
$${\left(\frac{\partial p}{\partial V}\right)}_{T}\to 0$$ as most of the heat added is converted into work of expansion. 

An experiment in the two-phase region, in a single cell that measures reversible heat added with increments of *T*, i.e., *Q*_rev_/Δ*T*, therefore, requires the definition of a heat capacity at saturation of gas or liquid, which is usually designated *C*_σ_. This can be calculated from the heat capacity *C_p_* if the variation in thermal pressure *γ*_σ_ = (d*p*/d*T*)_σ_ along the coexistence line is known. *C*_σ_(liq) and *C*_σ_(gas) are defined as the heat to reversibly increase the temperature of that phase in coexistence. *C*_σ_ for gas or liquid can be expressed in terms of available properties *C_p_*, *α**_p_* and *γ*_σ_: (8)$$C_{\sigma }=C_{p}+TV\alpha_{p}\gamma_{\sigma },$$
where *α**_p_* is the thermal expansivity, defined by:
$$\alpha_{p}=\frac{1}{V}{\left(\frac{\partial V}{\partial T}\right)}_{p}.$$

Thus, in order to interpret 2-phase isochoric heat capacity experiments [56,57,58], and the NASA space shuttle microgravity experiments [60], a fourth heat capacity has been defined by the lever rule [60,61] and designated *C*_λ_,
(9)$$C_{\lambda }=xC_{\sigma }(liq)+(1-x)C_{\sigma }(gas)+\Delta H_{v}{\left(\frac{\left|\partial x\right|}{\partial T}\right)}_{\sigma }$$
where Δ*H_v_* is the latent heat of evaporation. *C*_σ_ for both liquid and gas, and enthalpies of coexisting phases, and hence *C*_λ_, can be obtained for most pure atomic and molecular fluids from the NIST fluid property data bank [4]. Thermal expansivities in Equation (8) are calculated from Joule–Thompson coefficients: (10)$$\mu_{J-T}={\left(\frac{\partial p}{\partial T}\right)}_{H}=\frac{V\left(T\alpha_{p}-1\right)}{C_{p}}$$

### 4.3. Near Critical C_v_ Measurements

The heat capacity data used as evidence for a singularity at *T*_c_ with universal scaling properties [55,62], are reproduced in Figure 14. The values of *C_p_*, *C_v_* and *C*_λ_ for argon at the hypothetical critical density referred to in reference [49] are also shown in Figure 11. The values of *C*_λ_ calculated using Equations (7) to (10) can be compared to “*C_v_*” data reproduced from the first divergent *C_v_* results published in 1963 [62]. Figure 13 explains this apparent divergence of *C_v_*. This evident measurement of *C*_λ_, which looks the same as a lambda-like transition for the near critical data, is then re-used in reference [48]. This observation of a lambda-like dependence was first made by Uhlenbeck [10], half a century ago, who described it as “surprising” but encouraging with regard to the embryonic concept of universality. 

The divergence of *C*_λ_(*T*) near *T*_c_ in references [56,57,58,63,64], and Figure 13, occurs because *C*_σ_(*T*) for both liquid and gas diverges when the rigidity (d*p*/d*ρ*)*_T_* goes to zero on either side of the critical divide at *T*_c_. Any experimental measurement of *C*_λ_(*T*) at density *ρ*_exp_ within the critical divide, i.e., at *T*_c_, *ρ*_gas_ < *ρ*_exp_ < *ρ*_liq_, should thus show a divergence. If the hypothetical coexistence parabola were to exist, however, with a van der Waals singular point, the divergence would only occur at the hypothetical critical density *ρ*_c_. The isochoric measurements on argon [63,64] were performed at the densities of 521.0 kg/m^3^ and 531.0 kg/m^3^ [60], whereas the critical density given in NIST 2017 is *ρ*_c_ = 535.6 kg/m^3^ [4]. The results for a heat capacity divergence are supportive of the critical divide picture [7,8,9] and are inconsistent with the concept of a scaling singular point on the density, where a distinction between liquid and gas is deemed to disappear.

The comparisons of the calculated heat capacities with the experiments in Figure 13 are further vitiated as the experimental states investigated were homogenized by stirring to some degree. There is no basis to assume that the sub-critical measured steady-state heat capacity is equivalent to the thermodynamic equilibrium *C_v_*. Whilst stirring may reduce undesirable effects of gravitational phase separation, it creates an inhomogeneous shear field throughout the sample. For otherwise homogeneous Newtonian fluids with low viscosities this is normally acceptable for a single phase, e.g., in supercritical range. The thermodynamic states of these samples below *T*_c_, however, are in the 2-phase region. The stirred steady state is effectively a micro-emulsion. The surface energy of the emulsification contributes to the heat capacity measurements but is difficult to quantify. Below *T*_c_ the liquid-vapour surface tension is positive; it becomes zero at *T*_c_. The is no thermodynamic definition of a surface tension other than for two phases in coexistence; we can only speculate regarding the contribution of interfacial effects in the supercritical 2-state mesophases. 

It appears that the surface energy effect on heat capacity measurement cancels to some degree the enthalpy of evaporation (liq → vap) given by Δ*H_v_* in Equation (5) below *T*_c_. The surface work is of opposite sign, so the stirred heat capacities in the vicinity of *T*_c_ are in-between *C*_λ_ and *C_v_* with a weak divergence at *T*_c_ as seen in Figure 14. The pointed peak at around 40 J/K·mol in the isochoric heat capacity at *T_c_* is explained by an increase in surface tension and heterophase fluctuations starting around 140 K in the subcritical 2-phase range as Tc is approached. For *T* > *T_c_*, *C_v_* returns to the lower liquid-state value on exiting the mesophase at higher temperatures around 175K (see Figure 11). 

### 4.4. Space Shuttle Experiments

This also applies to the micro-gravity experimental measurement of a heat capacity for SF_6_ along a near critical isochore aboard the NASA space shuttle [59]. The publicly available data points of the NASA Space shuttle experiment are shown in Figure 15 alongside the heat capacities *C_p_*, *C_v_* and *C*_λ_ [4]. A comparison shows that what was actually measured in the space-shuttle experiment (labelled *C*_ss_) relates to *C*_λ_ a few degrees below *T*_c_, and to *C_v_* a few degrees above *T*_c_, but is possibly made to look like a lambda-transition in the vicinity of *T*_c_ by the stirring or steady-state emulsification. 

One of the consequences of the *ad hoc* acceptance of the van der Waals continuity hypothesis as scientific truism over the last 50 years is that many parameterizations of experimental measurements of thermodynamic properties have been prejudiced by the *a priori* assumption of a singularity that does not exist. The adoption of a singular point scaling parabola to interpolate the data between *ρ*_c_(gas) and *ρ*_c_(liq) at *T*_c_, with an intermediate singularity, is widespread and misleading. It leads to unreal values in the vicinity of *T*_c_. An example in the present context is the misinterpretation of SF_6_ space shuttle data. The real thermodynamic *C_v_* data for SF_6_ are the NIST values. An equation-of-state proposed by Kostrowicka-Wyczalkowska and Sengers [65] assumes a scaling singularity at the outset, and uses a “cross-over” mathematical device in order to accommodate the spurious divergent form of the experimental space shuttle measurements, although these data are not the thermodynamic equilibrium *C_v_*, defined by the lever rule as illustrated here in Figure 15. 

### 4.5. Heat Capacities from V(p,T)

Isochoric heat capacities (*C_v_*) derived from *p*-*V*-*T* equation-of-state measurements by the Michels group of the van der Waals Laboratory in Amsterdam, published in 1958 [61,62], are also widely used as evidence of continuity with a scaling singularity. The data in reference [54] (Table XXIV, p.793) suggest just the opposite conclusion: it shows no evidence for any critical point. The highest coexisting temperature reported [61,62] at which liquid and gas coexist of −122.5 °C (= 150.7 K) is right on the critical temperature (150.87 ± 0.015K [4]). At this temperature Michels et al. find a maximum coexisting gas density of 258.13 amagat (= 461.0 kg/m^3^) and a minimum coexisting liquid density of 343.25 amagat (= 613.2 kg/m^3^). These limiting coexistence densities compare well with the argon gas and liquid densities (475.6 and 598.6 kg/m^3^ respectively) obtained from the intersection of percolation loci used to define *T*_c_ and the coexisting state bounds *ρ*_c_(gas) and *ρ*_c_(liq) at *T*_c_ thermodynamically [8]. In addition, the isochoric heat capacities of these two state points at *T_c_*, *C_v_* (gas) and *C_v_* (liquid), give a mean *C_v_* (lever-rule) value, which is near the NIST subcritical and supercritical values to within ± 0.1 K, as shown in Figure 14.

## 5. Conclusions

From the foregoing review, a foremost conclusion is that the historic thermodynamic results from the van der Waals Laboratory are in fact inconsistent with van der Waals hypothesis of continuity of liquid and gas. They are also inconsistent with the existence of a critical point singularity of the density surface as implied in van der Waals equation-of-state [3]. The results of Michels et al. actually support the present approach to criticality, based upon the intersection of percolation loci [6,7,8], and a supercritical mesophase. Their results are consistent with a hiatus between liquid and gaseous states along supercritical isotherms and inconsistent with a divergent isochoric heat capacity at *T*_c_. 

There are no conclusions or scientific discussion of the results in the research papers of Michels et al. [59,61], even though there is clear evidence of a critical divide that contradicts the continuity hypothesis [2,3]. We have seen from the foregoing review, however, that, at the time of Michels-group research, there was already a substantial literature of compelling experimental evidence for non-continuity of gas and liquid along critical and supercritical isotherms. In 1955 the in-depth review article by O. K. Rice [48] cited no less than 25 literature references of experimental results contradicting van der Waals continuity hypothesis. The thermodynamic measurement results of Michels et al. [59,61] actually support the claims of Traube [32], who queried the gas-liquid continuity hypothesis of van der Waals citing a range of unequivocal experimental results as evidence of non-continuity. Rice [48] further outlined convincing theoretical arguments for a critical divide based upon divergent cluster integrals in the Mayer virial expansion of the partition function of a model molecular Hamiltonian. 

Our review concludes that Mayer’s original ideas accord with both the historic experimental literature of thermodynamic properties: divergent cluster sizes in the configurational integral equate with percolation bounds, consistent with the results from modern computer experiments [19,20,21,22,23,24,25,26,27]. Chapter 14 (“Condensation and the Critical Region”) [66] of the 1st Edition of Mayer and Mayer “Statistical Mechanics” (1940) is compelling evidence in the present context. Although Mayer did not have access to our present understanding of cluster percolation, his intuitive conclusions and background science are essentially correct, in relating divergence of cluster sizes to percolation transitions or 3^rd^ order phase transitions [67]. Unfortunately, by the time the 2nd Edition of Mayer and Mayer’s “Statistical Mechanics” was published, 20 years later, Joseph Mayer was no longer active in research, and the concept of universality had been acclaimed as established physics [10,11,12,13]. The original chapter on “Condensation and the Critical Region” is missing from the 2nd Edition of Mayer and Mayer.

The theory of van der Waals and his renowned equation-of-state survived for about 50 years until, increasingly, there was evidence that it could not account for the experimental data unless it was generalised to include more and more terms with many more adjustable parameters. Whilst modern equations-of-state with countless terms and parameters [4,68] can reproduce experimental *p*-*V*-*T* data to high precision, they tell us virtually nothing about the underlying physical science. The ultimate test of any scientific theory is its ability to account for experimental observations. From the foregoing review, we see that when the van der Waals hypothesis and the concept of universality is abandoned, the pressure equation-of-state, and hence also the thermodynamic state functions, can be expressed simply in terms of a few physical constants belonging to the fluid, and coefficients in virial expansions [28]. 

Surface tension plays a central role in the thermodynamic stabilisation of colloidal equilibrium states. Computer experiments on a model Lennard-Jones fluid show that it goes to zero at a finite density difference at *T*_c_ [25,27]. Surface tension, nevertheless, is the clue to understanding the thermodynamic states both above and below the critical divide. It is positive and well-defined below *T*_c_, and we have two-phase coexistence with liquid condensation that minimizes the interfacial area and hence also total Gibbs energy. At *T*_c_, the surface tension is zero, but it cannot be thermodynamically well-defined above T_c_ as the two states are not coexisting phases, but colloidally dispersed states that do not exist independently of each other, as the precursor states to the percolation transitions are dispersed hetero-phase fluctuations [69]. 

As temperature increases above T_c_ within the mesophase, the pressure difference between liquid and gas states along an isotherm at the percolation bounds increases. Equilibrium is obtained if both the liquid and gas pure phases, both inter-disperse to percolate the phase volume, with an interfacial surface tension that balances any Gibbs energy difference between gas and liquid at the same *T*,*p* state point. Thus, both pure gas and pure liquid states coexist at the same *T*,*p* states with uniform chemical potential throughout at equilibrium in the mesophase. Looking at the experimental data, (Figure 9, Figure 10 and Figure 11) it appears that the mesophase may extend all the way to low density at the Boyle temperature, narrowing the density gap as it does so. 

Comparing critical point hypothesis with experiment establishes a scientific truth. The data described as “experiment” in reference [4], i.e., as obtained from NIST tabulations, are based upon the TSW-equation-of-state [68] with hypothetical “crossover equations” [65] near T_c_. In the region of the critical divide and the immediate supercritical region, data points are either fabricated by hypothetical scaling equations, or distorted by the continuity hypothesis implicit in the equations-of-state used. These multiparameter equations tell us nothing about the underlying science of critical and supercritical state bounds; they produce hypothetical near-critical and supercritical mesophase data that have never been experimentally measured. Figure 16 shows how the scientific malpractice of assuming the van der Waals and universal scaling hypotheses to be established scientific truth, leads to the spurious experimental *p*-*V*-*T* data [4,19] in the vicinity of *T_c_*.

Finally, it only takes one experimental property to disagree with theory for the basic hypothesis to be wrong. The present equation-of-state analysis with confirmation of a critical divide would appear to suffice. However, there will always be those who cite the “logarithmic divergence of the isochoric heat capacity on both sides of *T*_c_” as discovered by the historic measurements of the 1960s and confirmed by the space shuttle microgravity experiment. It seems rather unusual, if these measurements were such an important fundamental cornerstone of the theory of critical-point universality [10,11,53], that for several decades, nobody has attempted to reproduce a divergent *C_v_*. It is also rather curious that the space-shuttle measurements in microgravity, designed to eliminate gravitational phase separation effects, found almost identical behaviour to the existing mundane laboratory measurements, i.e., a quasi-vertical increase in *C_v_*(*T*) for *T* ≥ *T*_c_. We conclude that there is a need for further direct accurate measurements of *T* > *T*_c_ near-critical heat capacities, not least to characterise the properties of the supercritical mesophase, and to investigate and characterize the effect of surface tension terms that must be present but difficult to define and quantify. 

In summary, we cannot find any reproducible scientific experimental evidence purporting to directly observe a critical singularity on the density surface, other than [56,57,58,59,60,61,62,63,64], which reaffirm the inferences of a critical divide and a supercritical mesophase [7,8,9]. We must conclude, therefore, that the non-continuity description of liquid-gas criticality, with a critical divide and a supercritical mesophase bounded by percolation loci, and with the properties of a colloidal co-existence, is the only plausible description so far proposed that is consistent with the results of 150 years of experimental thermodynamic measurement research. The vast body of experimental evidence suggests that the critical divide and supercritical mesophase are general properties belonging to all pure atomic and molecular fluids. 

## Figures and Tables

**Figure 1 entropy-22-00437-f001:**
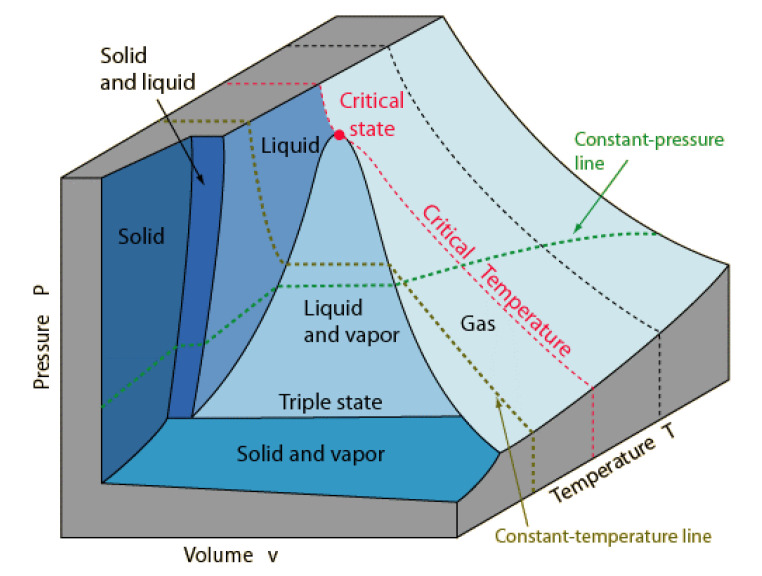
Illustration of a Gibbs (1873) surface: *p*(*V*,*T*) for a typical one-component pure fluid showing a hypothetical critical volume postulated by van der Waals (1873).

**Figure 2 entropy-22-00437-f002:**
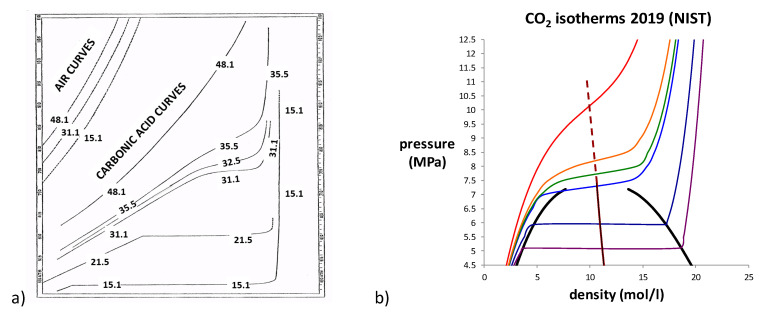
(**a**) The original pressure vs. density experimental data for 6 isotherms of CO_2_ in the region of the critical temperature (30.9 °C [4]) as reported by Andrews in 1869 [2]; the unlabelled abscissa scale is a logarithmic density; (**b**) the same 6 isotherms on a linear density scale from NIST [4]; also shown are the coexistence envelope (solid black lines) and the law of rectilinear diameters (almost vertical brown straight line) extended into the supercritical region (dashed).

**Figure 3 entropy-22-00437-f003:**
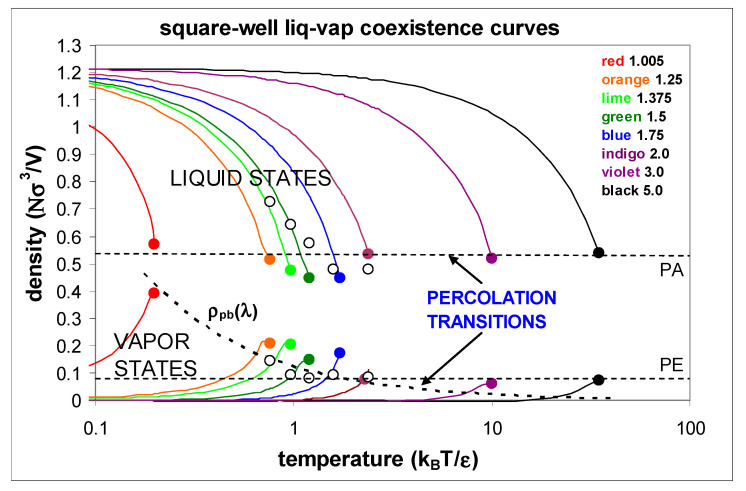
Coexistence curves of square-well fluids from references [6,7,19,20,21,22,23,24] with empirical limiting densities shown as closed circles and curves for *λ* = 1.25–2.0 from Vega et al. [20]; open circles are the densities of Elliott and Hu [21]; densities at l = 3.0 from Benavides et al. [22]; densities and curves for *λ* = 1.005 and *λ* = 5 [6,7]; the upper dashed line is the hard-sphere (HS) available volume percolation transition loci and the lower horizontal dashed lines are the extended volume percolation transition loci, of the hard-sphere reference fluid [6,7]; *ρ*_PB_(*T*) is the high-*T* limit, *λ*-dependent, bonded-cluster percolation density at *T_c_* of square-well fluids.

**Figure 4 entropy-22-00437-f004:**
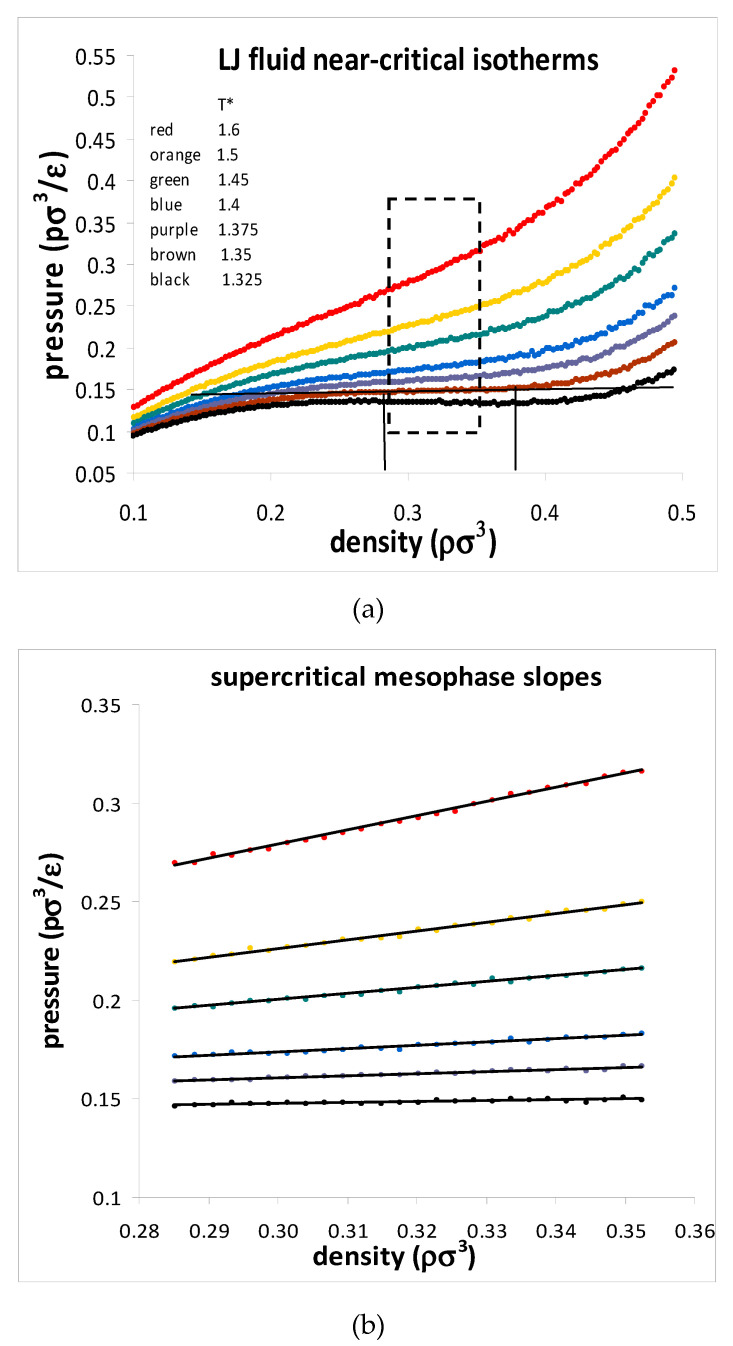
Critical and supercritical pressure isotherms of the Lennard–Jones fluid from reference [25]. (**a**) Isotherms showing the linear mesophase regions (dashed rectangle) between gas-like and liquid-like states of decreasing and increasing rigidities. (**b**) Expanded dashed rectangle region in (**a**) for 50 state points along each isotherm within the mesophase.

**Figure 5 entropy-22-00437-f005:**
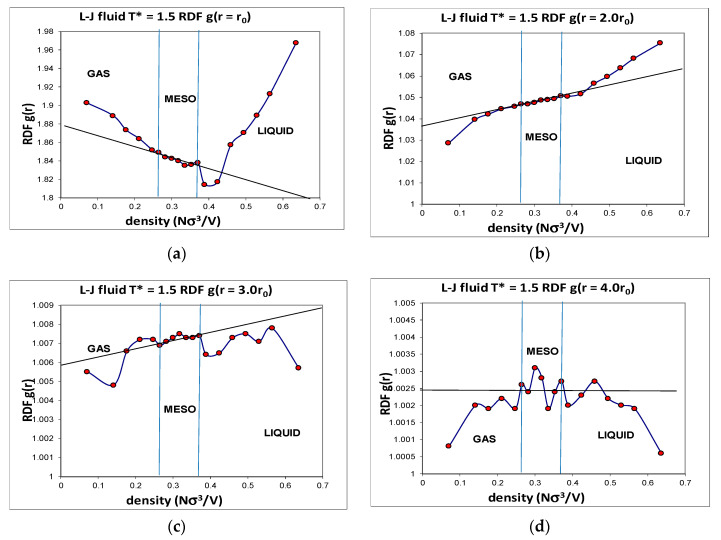
Values of the normalized pair distribution probability function at selected intermolecular pair distances ranging *r*_0_ to 4*r*_0_ (**a**–**d**); *r*_0_ (=2^1/6^*σ*) is the distance of zero force; the state parameters along the supercritical isotherm (*T** = 1.35); percolation transitions at *T_c_** = 1.336 are at reduced densities 0.266 (PB) and 0.376 (accessible volume (PA)) [25], as indicated by vertical lines.

**Figure 6 entropy-22-00437-f006:**
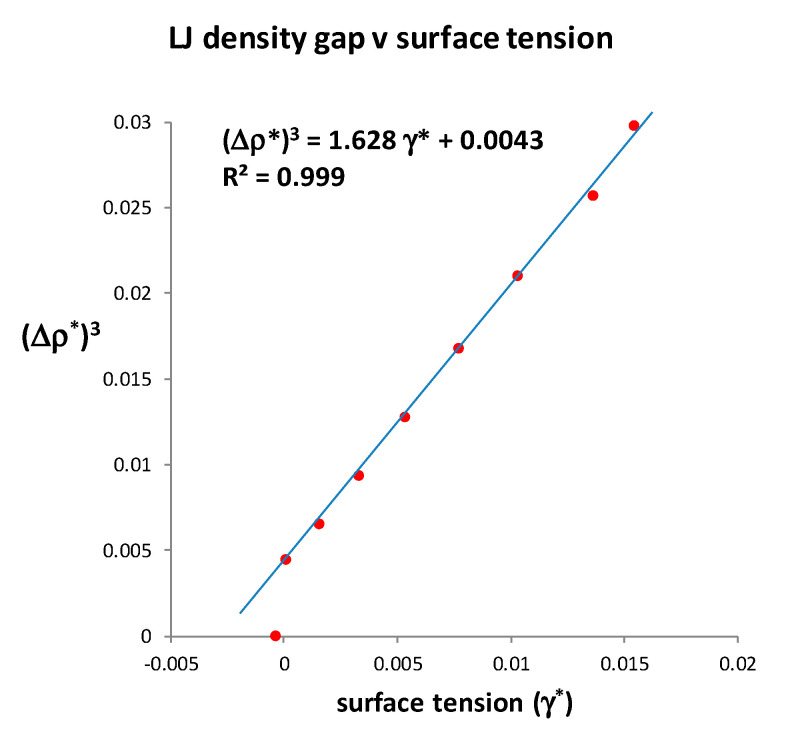
Near critical vapour-liquid density-difference dependence upon surface tension; *γ** goes to zero at a finite density difference, which coincides with the density difference between PA and PB at *T*_c_. The data points shown are taken directly from numerical results in Table 1 of the paper by Potoff and Panagiotopoulos [27].

**Figure 7 entropy-22-00437-f007:**
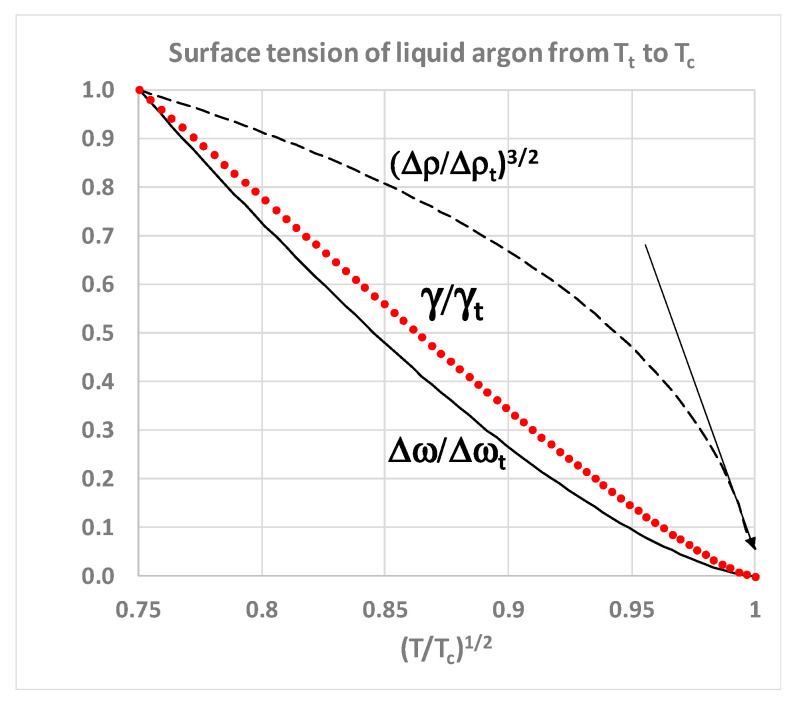
Surface tension (*γ*) of saturated liquid argon (red dots) obtained from NIST tables [4] relative to the triple point value (*γ*_t_) as a function of (*T*/*T*_c_)^1/2^. Also shown are the density difference (Δ*ρ*) ratio (dashed black line) and the rigidity difference (Δ*ω*) ratio (solid line) for comparison; the arrow shows that a real liquid argon shows a cubic density-difference dependence on *T* in the proximity of *T*_c_, consistent with the reinterpretation of the MC results in Figure 6.

**Figure 8 entropy-22-00437-f008:**
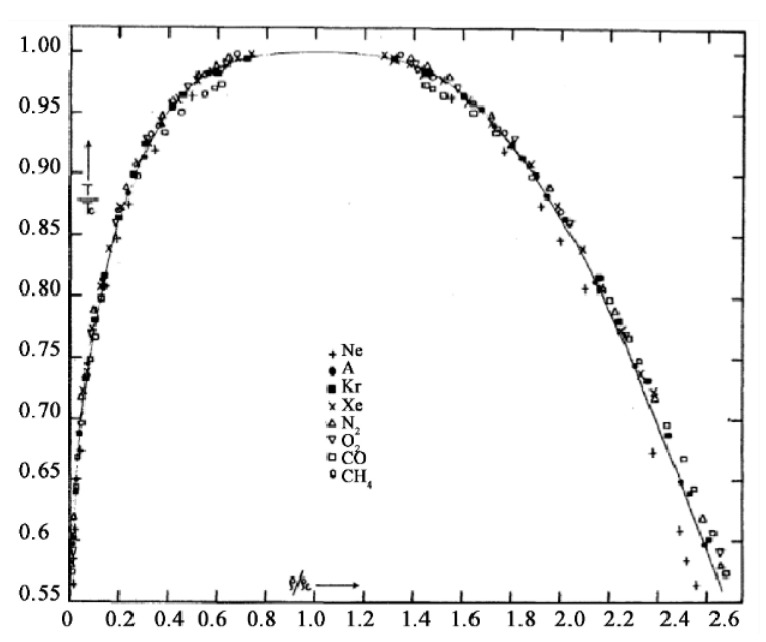
Guggenheim’s classic diagram of 1945 [42]. A plot of reduced temperature (*T*/*T*_c_) as a function of reduced density (*ρ*/*ρ*_c_) of available experimental data points for the coexisting densities of 8 fluids: the solid line is an assumed parabola with a singularity of the form *ρ* − *ρ*_c_ = (*T* − *T*_c_)^1/3^ to determine a critical density.

**Figure 9 entropy-22-00437-f009:**
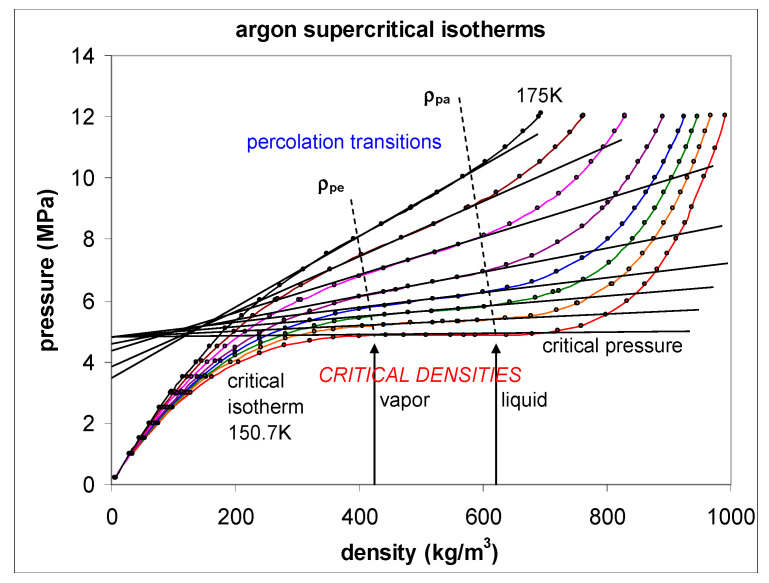
Experimental pressure measurements along near-critical isotherms plotted directly from original measurements as reported by Gilgen et al. [50,51]. The black lines have been added to illuminate the linear mesophase region, including the density gap along the critical isotherm.

**Figure 10 entropy-22-00437-f010:**
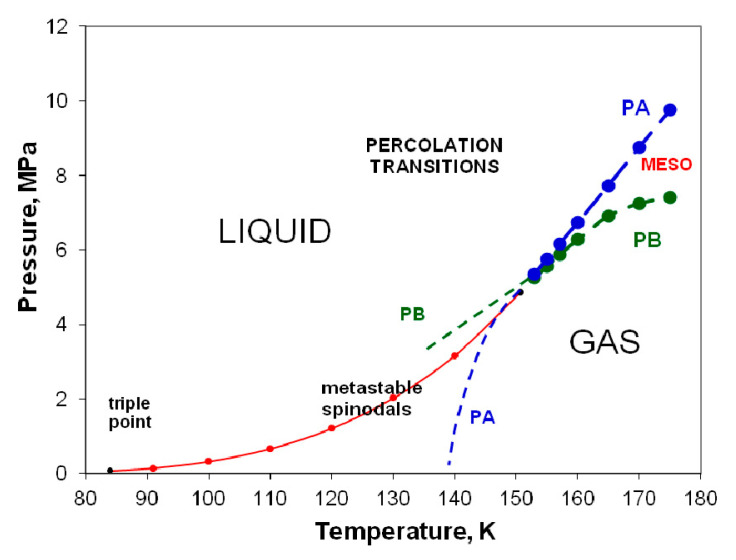
Percolation transition points along near-critical isotherms plotted directly from original measurements as reported by Gilgen et al. [50] and shown in Figure 4 in the *p*-*T* projection. The extrapolated dashed lines of available volume (PA-blue) and bonded cluster (PB-green) correspond to the experimentally observed spinodal lines from the literature. The red line and points are the coexistence data of Gilgen et al. [51].

**Figure 11 entropy-22-00437-f011:**
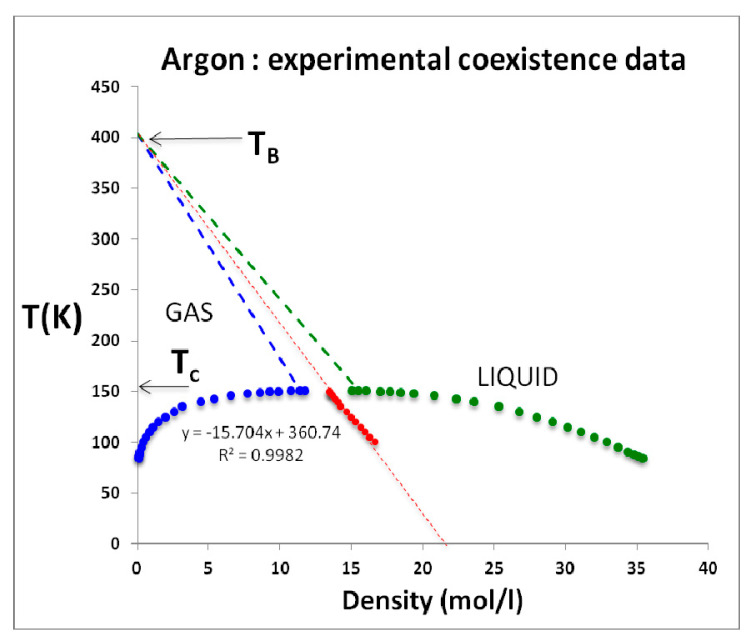
Experimental coexistence density measurements taken directly from Table 1 of Gilgen et al. [50]; at *T*_c_ the experimental density gap coincides with the linear density difference between the supercritical gas and liquid phases as defined by the sign of the rigidity state function and obtained from the NIST 2017 Thermophysical tables [4]; the percolation loci (dashed) decrease linearly with density in the supercritical range and appear to originate at the Boyle temperature; the large red data points are the LRD mean coexisting densities.

**Figure 12 entropy-22-00437-f012:**
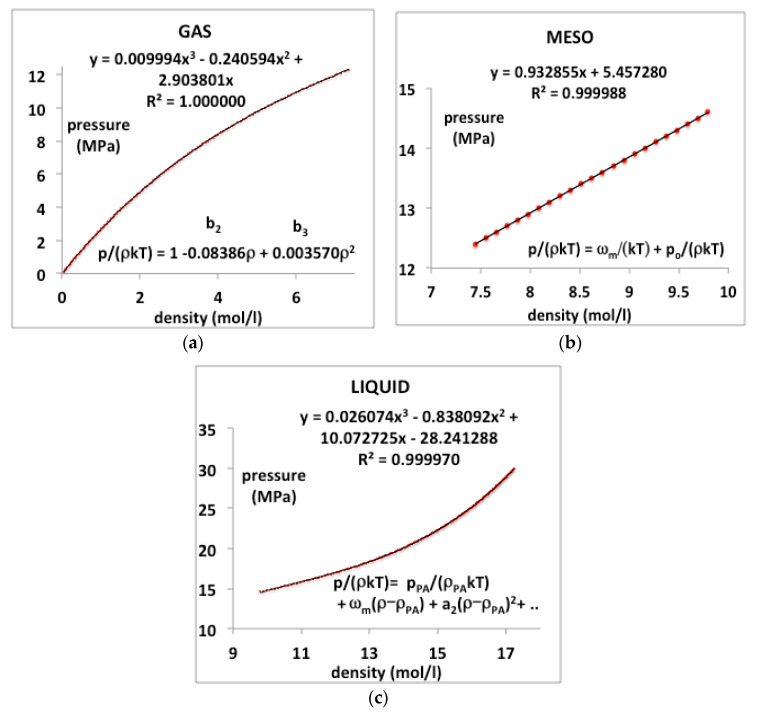
Pressure as a function of density for CO_2_ at a supercritical temperature (350 K or *T*/*T*_c_ = 1.15) as derived from NIST Thermophysical Properties compilation: (**a**) gas state, (**b**) supercritical meso-phase and (**c**) liquid.

**Figure 13 entropy-22-00437-f013:**
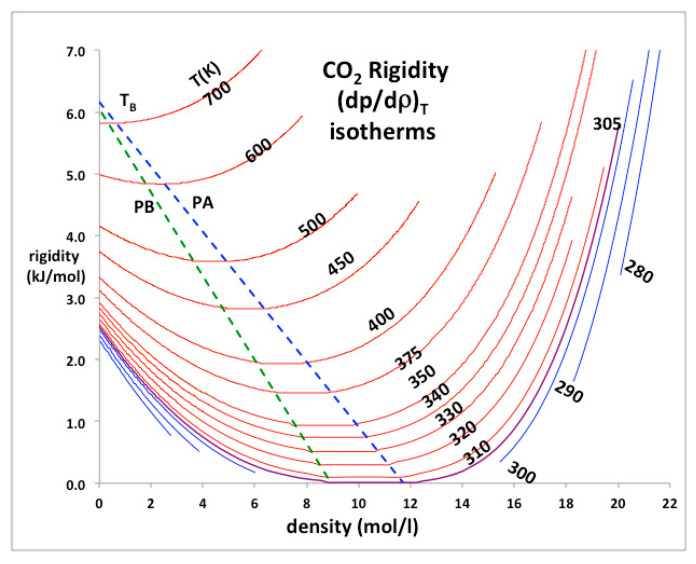
Supercritical isotherms of carbon dioxide rigidity: red = supercritical, purple = critical, blue = subcritical. The rigidity is obtained from NIST thermophysical tables [4]. Loci of the gas and liquid supercritical phase bounds are green and blue, respectively, and appear to merge at or near to the Boyle temperature (*T*_B_ ~ 700 K).

**Figure 14 entropy-22-00437-f014:**
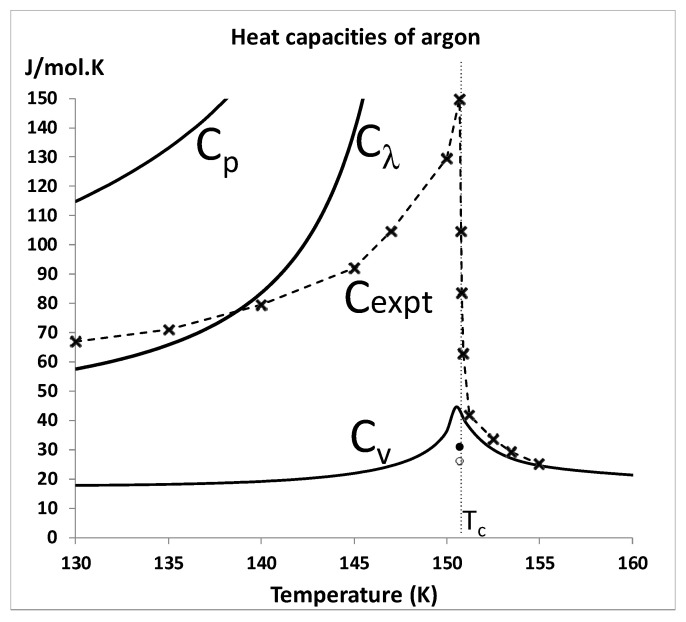
Heat capacities of argon for the isochore 13.29 mol/l. *C*_l_ is calculated from the NIST thermophysical compilations [4] using Equations (7) to (10); *C_p_* and *C_v_* are the NIST values; the two experimental *C_v_* data points at *T*_c_ are the values of *C_v_* at the densities of coexisting liquid (open circle) and gas (solid circle) taken from Table XX p.789 of Michels et al. [62]; the crosses are experimental data points reported and discussed in references [55,58,60]. (note that values for *C_p_* and C_λ_ above *T*_c_ diverge at *T*_c_ off the scale shown).

**Figure 15 entropy-22-00437-f015:**
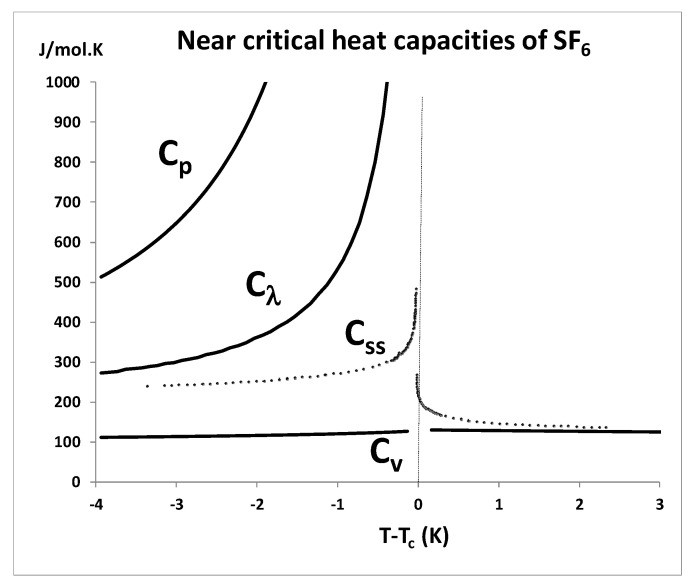
Variation of the heat capacities of SF_6_ along an isochore (5.0474 mol/l) within the critical divide, corresponding to the same density of the space shuttle experiment [59]. This density is about 1% lower than the mean critical density given by NIST [4] i.e., 5.0926 mol/L. *C*_ss_ are the space shuttle experimental data points; the subcritical and supercritical isochoric heat capacities (*C*_v_) are from NIST thermophysical compilations [4].

**Figure 16 entropy-22-00437-f016:**
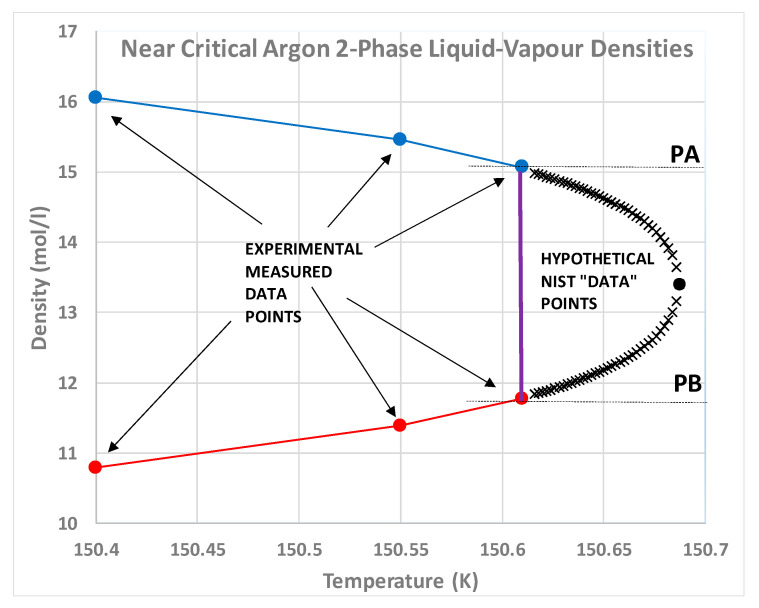
Near-critical density coexistence data for argon. The blue and red dots are experimental data points up to the maximum experimentally observed coexisting gas density and minimum coexisting liquid densities reported by Gilgen et al. [51]; the blue and red experimental coexistence lines up to the critical hiatus (purple line) are NIST data; the black crosses are also NIST data points [4] fabricated using equations that accommodate the continuity and universality singular critical density hypotheses; dotted lines show the percolation lines PA and PB that bound the supercritical mesophase as defined at a *T_c_* by the coexisting densities at equal pressures; there are no known experimental *p*-*V*-*T* data to support the hypothetical van der Waals critical point of argon as published by NIST [4] (black spot).

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
