# Peer review of "Supercritical Fluid Gaseous and Liquid States: A Review of Experimental Results"

_entropy, 2020, doi:10.3390/e22040437_

Round 1

Reviewer 1 Report

In this work, the authors review the available experimental evidence for non-existence of a critical point singularity on the Gibbs density surface. The authors consider the experimental evidence from the Andrews CO2 experiments until nowadays data included in authorized databases, such as NIST and Detherm, and computer results in order to provide evidence relates to the misunderstanding of critical point singularity as well as the concept of universality. The authors attribute this misunderstanding due to a misinterpretation of the isochoric hear capacity.

This topic has been captured the attention of one of the authors for years, who provide a very complete review and collection of several evidence that support the high quality and well writing review reported here.

Due to the relevance of this topic for the thermodynamic community, this review should be accepted in this journal.

Some minor comments of the review is related to the surface tension, this topic is not explored in this review in details but will be clarify or reaffirm some ideas reported in this work. Therefore, it is advice to consider this topic in depth for a future exploration of critical point singularity as well as the concept of universality

Author Response

we enclose our response in the attached file

Reviewer 2 Report

The authors present a very interesting manuscript. They have a discussion about the non-existence of a critical-point singularity on Gibbs density surface, for the existence of a critical dividing line between 2-phase coexistence, for a supercritical mesophase with the colloidal characteristics of a one-component 2-state phase, and for the percolation loci that bound the existence of gaseous and liquid states. The authors conclude that the body of extensive scientific experimental evidence has never supported the Andrews-van der Waals theory of continuity of liquid and gas, or the existence of a singular critical point with universal scaling properties. The manuscript seems to be correct and can be published, however, the authors should consider the following comments.   1. It is known that critical compressibility is highly difficult to trap with equations of state due to their orders of magnitude, what would be the authors' contribution to this problem?   2. In Figure 2 label, the authors mention "....CO2 in the region of the critical temperature (30.9K[4]) as...." This information is right?   3. Why not study more compounds than those presented and go deeper into the subject?
  4. What the authors mean by "There is evidence from the literature that it cannot be measured in a laboratory with accuracy in the vicinity of Tc".  What contribution can surface tension have?   5. It is known that compressibility varies with the size of the molecule, how do the authors consider this effect?
  6. The effects of molecule geometry and size, have they been considered?
  7. What is presented by the authors, can be interpreted as a generality?     As mentioned above, I recommend that the manuscript be published after major corrections noted.

Author Response

we enclose our replies in the attached file

Round 2

Reviewer 2 Report

The authors have complied with all the corrections and comments made by the reviewer. Therefore, I agree with the publication of this manuscript